# Amnion as an Innovative Antiseptic Carrier: A Comparison of the Efficacy of Allogeneic and Xenogeneic Transplantations in the Context of Burn Therapy

**DOI:** 10.3390/medicina60061015

**Published:** 2024-06-20

**Authors:** Agnieszka Klama-Baryła, Anna Sitkowska, Wojciech Łabuś, Przemysław Strzelec, Małgorzata Kraut, Wojciech Smętek, Wojciech Śliwiński, Ryszard Maciejowski, Marcin Gierek

**Affiliations:** 1Stanisław Sakiel Burn Treatment Centre in Siemianowice Śląskie, 2 Jana Pawła II Street, 41-100 Siemianowice Śląskie, Poland; anna.sitkowska@clo.com.pl (A.S.); wojciech.labus@clo.com.pl (W.Ł.); przemyslaw.strzelec@clo.com.pl (P.S.); gosia.kraut@clo.com.pl (M.K.); smetekwoj@gmail.com (W.S.); wojciechsliwinski@wp.pl (W.Ś.); ryszardmaciejowski@wp.pl (R.M.); marcin.gierek@clo.com.pl (M.G.); 2Faculty of Management, Warsaw University of Technology, 85 Narbutta Street, 02-524 Warsaw, Poland

**Keywords:** amniotic membrane grafts, antiseptics, burns

## Abstract

*Background and Objectives:* The amniotic membrane is widely used in the treatment of chronic wounds, in toxic epidermal necrolysis (TEN), and in the treatment of burns. In our clinical practice, we use amniotic dressings on shallow skin wounds caused by burns. Counteracting infections is an important aspect of working with burn wounds. Therefore, the main goals of this work are to demonstrate the usefulness of amniotic membrane soaked in antiseptics for the prevention of wound infections and to compare the antibacterial efficacy of selected variants of allogeneic and xenogeneic amniotic membrane grafts soaked in specific antiseptic agents. *Materials and Methods:* The studied material consisted of human and pig placenta. The human and animal amnions were divided in two parts. The first part consisted of amniotic discs placed on rigid mesh discs and preparing the fresh amnion. The second part of the amnion was frozen at a temperature of −80 °C for 24 h. Then, it was radio-sterilized with a dose of 35 kGy. The amniotic discs were placed on rigid mesh to prepare the radiation-sterilized amnion. The amniotic discs were placed in a 12-well plate and immersed in 3 mL of the appropriate antiseptic solutions: Prontosan, Braunol, Borasol, Microdacyn, Octenilin, Sutrisept, and NaCl as a control. The amniotic discs were incubated in antiseptics for 3 h. The microbiological tests were conducted by placing the antiseptic-infused amniotic discs on microbiological media inoculated with hospital strains. *Results:* The largest average zone of growth inhibition was observed in dressings soaked with Sutrisept, Braunol, and Prontosan. The greatest inhibition of bacterial growth was achieved for radiation-sterilized porcine amnion impregnated with Braunol and Sutrisept, as well as for radiation-sterilized human amnion impregnated with Braunol. *Conclusions:* Human and porcine amniotic membrane is effective in carrying antiseptics. Radiation-sterilized amnion seems to inhibit the growth of microorganisms better than fresh amnion.

## 1. Introduction

The amniotic membrane constitutes the inner layer of the placenta and performs a range of physiological functions (including providing the proper environment for the fetus, nourishing and protecting the fetus, as well as acting as an immunologically privileged barrier during embryo development) [1,2]. Its natural origin is a significant advantage due to the ease of obtaining the placenta both in humans and in animals. Fetal membranes constitute biological waste that can be collected during childbirth. The human placenta is obtained by gynecologists during a caesarean section or natural labor from patients who have expressed informed consent [3,4,5].

In our clinical practice, the amniotic membrane is used to treat shallow skin defects. Due to the location and aesthetic aspects, covering shallow burn wounds on the face provides very good clinical outcomes. Facial burns pose a serious problem among both adult and pediatric patients. Primarily for aesthetic reasons, ensuring the proper healing of shallow burn wounds is crucial for achieving the desired cosmetic effect. Years of clinical experience has also demonstrated the efficacy of amnion in the local treatment of toxic epidermal necrolysis (TEN) [6]. Toxic epidermal necrolysis, also known as Lyell’s syndrome, is a rare but life-threatening mucocutaneous disorder with an epidermal detachment of a total body surface area (TBSA) of >30% [7]. It often appears as a serious adverse reaction to medication and less frequently as a complication of skin infections. Due to the high risk of mortality, the treatment of patients requires prompt diagnosis, the identification and discontinuation of the causative agent, as well as specialized supportive care [8]. The prompt withdrawal of the suspected drug, fluid and electrolyte replacement, and topical wound care are the first lines of therapy [9]. Although systemic interventions may alter the clinical course of these conditions, adjunctive local treatment contributes to increased survival and accelerates wound healing [10]. Nevertheless, from a clinical perspective, wound infection should not be allowed, and in the event of its occurrence, the application of topical antiseptics may be a decisive treatment for these patients [11].

Both in shallow burns of body areas susceptible to stress (including the face and groin regions) and in toxic epidermal necrolysis (TEN), amnion dressings are increasingly used for topical treatment [12]. This is supported by the lack of expression of histocompatibility antigens in the amnion; hence, its transplantation is possible without the need for immunosuppression and the risk of graft rejection. The amnion demonstrates anti-inflammatory properties associated with the presence of anti-inflammatory mediators and the ability to induce the apoptosis of mononuclear cells, including lymphocytes and macrophages. Furthermore, it reduces the expression of major histocompatibility complex (MHC) class II. Its anti-inflammatory effect of the amnion indirectly prevents scar formation [3,13].

Its antibacterial properties also support its widespread clinical use, and they have a significantly lower risk of postoperative complications [3,4,5]. The problem of wound infection is a serious complication, often preventing the proper healing of skin defects. The use of locally applied amnion impregnated with antiseptics opens up new possibilities for the treatment of burns and its application in other clinical indications [2,9].

Due to the deficit of human organ and tissue transplant donors, the clinical application of grafts from alternative sources, such as those originating from animals, including transgenic ones, can be considered [14].

The primary goal of this study was to compare the antibacterial efficacy of selected variants of allogeneic and xenogeneic (transgenic pigs) amniotic membrane grafts soaked in specific antiseptic agents.

## 2. Materials and Methods

The studied material consisted of human placenta obtained during a natural birth, from which amniotic membrane grafts were prepared. Using a Biopsy Punch (Integra Miltex) with an 8 mm diameter, discs were cut from the human amniotic membrane in order to be impregnated with antiseptics.

The studied material also included pig placenta obtained during a natural birth. The pigs used in this study had previously undergone genetic modifications in accordance with the procedure described earlier [15]. Using a Biopsy Punch (Integra Miltex) with an 8 mm diameter, discs were cut from the porcine amniotic membrane in order to be impregnated with antiseptics.

### 2.1. The Reagents Used to Soak the Discs from the Human and Porcine Amniotic Membranes

The following reagents were used:Control NaCl 0.9% solution;Prontosan (polyhexanide 0.1%, betaine 0.1%) (manufacturer: B Braun Melsungen AG, Melsungen, Germany);Braunol 7.5% (100 g of solution contains 7.5 g of povidone iodine—Povidonum iodinatum with a 10% iodine content) (manufacturer: B Braun Melsungen AG, Germany);Borasol 30 mg/g (1 g of skin solution contains 30 mg of boric acid) (manufacturer: PROLAB, Naklon/Notecią, Poland);Microdacyn (40 ppm hypochlorous acid (HOCl)) (manufacturer: Oculus Technologies of México S.A. de C. V., Zapopan, Jalisco, Mexico);Octenilin (Aqua valde purificata, Glycerol, Ethylhexylglycerin, Octenidine HCl) (manufacturer: Schülke&Mayr GmbH, Warsaw, Poland);Sutrisept (0.1% PHMB—polyhexanide; 1% Poloxamer 188) (manufacturer: ACTO GmbH, Warsaw, Poland).

### 2.2. The Collection and Preparation of the Amnion, Followed by Its Incubation in Antiseptics

The procedure of obtaining the placenta during natural childbirth was conducted in accordance with the principles of asepsis. The obtained placenta was rinsed in a saline solution. After transferring the amnion into the laminar flow chamber, the amnion was manually separated from the chorionic layer. The amnion was again rinsed in the saline solution and divided into 2 parts depending on the stage of the experiment.

In the first stage, the freshly collected amnion was placed on a dental glass plate, and discs for soaking in antiseptics were cut out of the amniotic membrane using a Biopsy Punch (Integra Miltex) with a diameter of 8 mm. The amniotic discs were placed on rigid mesh discs of the same diameter for easier manipulation to prepare the fresh amnion (biologically vital or live).

The second part of the amniotic membrane was packed into double plastic bags, 1 mL of 0.9% NaCl was added, and the material was frozen at a temperature of −80 °C for 24 h. Then, it was radio-sterilized with a dose of 35 kGy. Subsequently, the amniotic membrane was thawed, and discs for soaking in antiseptics were cut out of the human amniotic membrane using a Biopsy Punch (Integra Miltex) with a diameter of 8 mm. The amniotic discs were placed on rigid mesh discs of the same diameter for easier manipulation to prepare the radiation-sterilized amnion (biostatic or dead).

The amniotic discs from each part (fresh and radiation-sterilized) were placed (with 3 repetitions for each antiseptic) in a 12-well plate and immersed in 3 mL of the appropriate antiseptic solutions: Prontosan, Braunol, Borasol, Microdacyn, Octenilin, Sutrisept, and NaCl as a control. The amniotic discs were incubated in antiseptics for 3 h. After incubation, they were sent for microbiological testing to assess the inhibition of microbial growth. The microbiological tests were conducted by placing the antiseptic-infused amniotic discs on microbiological media inoculated with hospital strains isolated from clinical material. The studied material was collected from the patients hospitalized in the Stanisław Sakiel Burn Treatment Center in Siemianowice Śląskie, who demonstrated clinical symptoms of bacterial infection (fever, wound appearance, and characteristic smell). The zones of inhibited growth of microorganisms were measured in millimeters.

### 2.3. Statistical Analysis

The statistical analysis was conducted using the STATISTICA 12 software. The normality of distribution was examined using the Shapiro–Wilk test. The hypotheses were tested using the Kruskal–Wallis test (to compare at least four groups) and Mann–Whitney U test (to compare two groups). The level of significance was set as 0.05 (5%). Due to the limited number of samples, only differences between human and porcine amnion and between different types of tissue preservation (fresh and radiation-sterilized), as well as between individual antiseptics, were compared.

## 3. Results

The microbiological tests were conducted by placing specific antiseptic-infused amniotic discs on microbiological media inoculated with hospital strains isolated from clinical material. The diameter of growth inhibition of *Pseudomonas aeruginosa* (Figure 1), *Acinetobacter baumannii* (Figure 2), and *Methicillin-sensitive Staphylococcus aureus* (Figure 3) was measured when using soaked fresh human amnion, radiation-sterilized human amnion, fresh porcine amnion, and radiation-sterilized porcine amnion.

In this study, the zones of growth inhibition for individual strains isolated and cultivated from clinical material were compared for fresh human amnion (biologically vital) and radiation-sterilized human amnion (biostatic) impregnated with antiseptics (Table 1 and Table 2).

The zone of growth inhibition for each strain that was isolated and cultivated from clinical material was compared for fresh porcine amnion soaked in antiseptics (Table 3) with porcine amnion sterilized with radiation (Table 4).

When analyzing the utility of human and animal amnion for clinical applications as well as fresh (biologically vital) and radiation-sterilized (biostatic) amnion, the zone of growth inhibition was compared for the antiseptic-soaked human amnion, the fresh and radiation-sterilized human amnion, and for the fresh and radiation-sterilized porcine amnion. The average zone of growth inhibition for the control—NaCl—is 8.8, and the average zone of growth inhibition for the applied antiseptics is 12.2 regardless of the tested material (Figure 4).

Statistically significant differences were also demonstrated between the applied antiseptic and the obtained zone of growth inhibition of the analyzed bacterial strains: *P.aeruginosa* (*p* = 0.013), *A. baumannii* (*p* = 0.023), and *S. aureus* (*p* = 0.003). The largest average zone of growth inhibition was observed in dressings soaked with Sutrisept (14.8) and Braunol (14.8) and Prontosan (12.3) (Figure 5). The greatest inhibition of bacterial growth was achieved for radiation-sterilized porcine amnion impregnated with Braunol (16 mm) and Sutrisept (16.3 mm), as well as for radiation-sterilized human amnion impregnated with Braunol (16.3 mm) (Figure 5).

Figure 6, Figure 7 and Figure 8 provide a box plot of the statistical data and displays the distribution within the hospital strains taken to the experiment.

## 4. Discussion

Our research indicates the potential for using human amniotic membrane in the treatment of chronic wounds, including infected wounds. Many studies describe the possibility of using the amnion graft alone in the treatment of wounds as it promotes the healing process [7,16,17]. Katie et al. identified several critical factors that may be crucial in the amnion’s stimulation of the wound healing process. These factors include the extracellular matrix (ECM) of the amnion, cytokines and growth factors, stem cells and their immunomodulatory properties, and the wound environment. The amniotic membrane is used with great clinical success in the treatment of chronic wounds of various etiologies [16]. The amniotic membrane was used in the treatment of chronic wounds in situations where traditional methods were not effective over a one-month treatment period. It was proven that treating wounds with the amniotic membrane significantly improves healing by reducing the wound area and shortening the time needed to complete healing [11,16]. A group of 117 patients were treated on an outpatient basis using amniotic membrane grafts. Diabetic foot ulcers were diagnosed in 34% of placental allograft recipients, venous ulcers of the limbs were present in 25% of recipients, surgical wounds were observed in 20% of recipients, bedsores were observed in 14% of recipients, ischemic wounds were observed in 6% of recipients, and 2% of the patients suffered from traumatic wounds. The retrospective analysis showed that complete healing occurred in 91.1% of the treated patients who were administered grafts 5 times weekly on average [7]. Human amniotic membrane was also used in the treatment of chronic infected burn wounds [17]. In a group of 38 patients, the most commonly isolated microorganism from the wound was Staphylococcus (62.85% of patients) [17]. These studies showed that the amniotic membrane significantly contributed to the acceptance of the graft on clean wounds, serving as external protection for the skin graft. The amniotic membrane can be used as an important dressing in the treatment of infected chronic wounds after burns due to its antibacterial properties [17,18,19,20,21,22,23]. In our study, we also demonstrated the ability of the amniotic membrane to inhibit the growth of bacteria isolated from burn wounds, including methicillin-susceptible Staphylococcus aureus (MSSA). The use of the amnion soaked in saline solution, regardless of the method of preparation, inhibits the growth of bacteria isolated from burn wounds at a constant minimum level (8 mm zone of bacterial growth inhibition).

There are also studies indicating the impact of the method of preparing the amnion on its properties in promoting wound healing. It was investigated whether conducting dehydration procedures or deep-freezing storage of the amnion affect the wound healing process. Lyophilized sterilized placental grafts were compared with hypothermically stored live amniotic membrane grafts [16]. These studies demonstrate the importance of amnion processing techniques in the wound healing process [11,16,17,24,25,26,27,28,29,30]. Our research also confirms the significance of the method of processing the amniotic membrane in inhibiting the growth of bacterial strains isolated from burn wounds. It was shown that when the amnion itself is soaked in saline solution, it inhibits bacterial growth to the same extent regardless of the method of preparation (8 mm zones of bacterial growth inhibition). However, we demonstrated that the method of preparing the amnion soaked in antiseptics has a significant impact on the degree of microbial inhibition. We showed that there was a much greater inhibition of microbial growth when the amnion was sterilized by radiation and previously subjected to a cryopreservation procedure and then soaked in antiseptics. Significant differences in inhibition were observed when inhibiting the growth of methicillin-susceptible *Staphylococcus aureus* (MSSA) using a cryopreserved and then radiation-sterilized amnion soaked in Sutrisept (25 mm zone of bacterial growth inhibition) compared to fresh and biologically vital amnion, i.e., live amnion soaked in this antiseptic (17 mm zone of bacterial growth inhibition). Almost all bacteria were more inhibited by the radiation-sterilized amnion soaked in antiseptics than the fresh (biologically vital) amnion. Only in the case of methicillin-sensitive *Staphylococcus aureus* (MSSA) growth inhibition by amnion soaked in Braunol was a larger zone of growth inhibition observed when using fresh amnion than radiation-sterilized amnion. The inhibition of *Acinetobacter baumannii* growth by amnion soaked in Borasol is also greater when using fresh amnion compared to radiation-sterilized amnion. For all other antiseptics (Prontosan, Octenilin, and Sutrisept), this relationship is reversed. These results indicate that radiation-sterilized amniotic membrane has better absorbent properties compared to fresh amnion. Our research also indicates weak inhibition properties of microbial growth isolated from burn wounds by amnion soaked in Microdacyn regardless of the method of preparation. The zones of growth inhibition of all tested microorganisms were the same compared with amnion soaked in saline solution (8 mm zone of bacterial growth inhibition). In the case of radiation-sterilized amnion, the best growth inhibition properties for methicillin-sensitive *Staphylococcus aureus* (MSSA) and *Pseudomonas aeruginosa* were demonstrated for Sutrisept, while for *Acinetobacter baumannii*, the best growth inhibition properties were observed for Braunol. Both antiseptics showed the best inhibition properties of growth of the tested bacteria regardless of the type of amnion used.

Placental transplants are often used as an alternative therapy when standard treatments do not yield the desired results. This is particularly true for infected wounds, where using the patient’s own grafts may lead to their loss due to infection. Amniotic membrane soaked in antiseptics can be used as a dressing to protect the patient’s own grafts, preventing infection. These transplants have several advantages over many other available bioactive therapies, including a low cost, ease of manipulation, low immunogenicity and antibacterial properties, modulation of inflammation, the ability to promote cell migration and proliferation, and the stimulation of stem cell activity. These cellular responses positively influence the wound healing process by accelerating tissue remodeling and inhibiting scarring. The low cost of preparing transplants and the lack of ethical dilemmas are leading to the increased clinical use of placental tissues.

Amniotic grafts are also characterized by high safety. Firstly, the procedures for their preparation are conducted in tissue and cell banks, which must meet the appropriate quality standards and obtain the necessary certifications, ensuring the safety, consistency, and quality of the transplants. Amniotic grafts are also low in immunogenicity, which ensures their immunological safety for patients. Due to the properties of the amniotic membrane, such as the lack of expression of tissue compatibility antigens, it can be transplanted without the need for immunosuppression and without the risk of transplant rejection. This is especially beneficial for patients with burns, where the developing burn disease leads to a weakened immune system. The use of low-immunogenicity amniotic grafts soaked in antiseptics should yield desired wound healing effects, both for infected wounds and as a preventative measure against infections.

The frequency of changing the antiseptic-soaked amniotic dressing should be adjusted to the clinical state of the wound and should continue until healing or control of the infection is achieved. Wounds often require an individualized treatment plan, which is also related to the type of microorganisms isolated from the wounds. Our study aims to indicate the possibilities of using this simple procedure of soaking amnion in hospital-available antiseptics in a clinical setting to ensure better availability and longer-lasting effects. The aforementioned properties of amnion in supporting wound healing are also significant. The use of amnion alone is not only a standard procedure for treating burn wounds, but it is also gaining popularity for application in other conditions.

Fetal tissues are obtained from women who are qualified based on negative serological test results and clinical interviews. Donors must give informed consent for the collection of fetal tissues. They decide to donate placentas to the tissue bank only when it does not pose a threat to their health or the health of the child. Placentas are typically obtained during cesarean sections, but it is possible to use a placenta delivered vaginally after conducting tests and completing all necessary qualification procedures. However, cesarean sections allow for the collection of tissue material under aseptic conditions without passing through the birth canal. Regardless of the delivery method, donors must be tested for infectious diseases, including HIV (human immunodeficiency virus), hepatitis B and C, and syphilis, in accordance with legal requirements.

The tissue must be sterile to allow for the preparation of biovital grafts that have not undergone radiation sterilization procedures. This represents a significant limitation in the availability of this type of graft and, consequently, in obtaining samples for research. These samples are waste products remaining after the preparation of the amnion; hence, the sample size in this study was limited and often could not be repeated with amnion from the same donor. The authors aimed to translate the research findings into clinical practice as much as possible. Therefore, strains isolated from patients’ wounds were used in this study. Bacteria were selected as they are the most common causes of hospital infections in burn centers, which often hinder progress in wound treatment for these patients.

The use of strains isolated from patients in this study also presents certain limitations in the ability to replicate this research. However, this is offset by the applicability of the results to direct clinical practice in burn centers, where the cultivated bacterial flora is quite characteristic, and these infections often pose a significant problem in patient treatment.

One useful function of the amnion is its absorption properties. The amnion can be soaked in a liquid and transplanted into a wound where it guarantees an appropriate level of humidity. Using the amnion as a medium for antibiotics or antiseptics and as a means of healing and preventing infections may become a revolutionary application of the amnion. There are publications that confirm the possibility of using the amnion as a medium for delivering drugs. Amniotic membrane, amniotic mesenchymal cells (AMCs), and amniotic epithelial cells (AECs) are often used in skin tissue engineering as a drug reservoir [31]. Human amniotic cells and the matrix of amniotic membranes have the ability to store drugs and release them in the body, which is clinically useful because they also show resistance to many drugs, e.g., paclitaxel (PTX) [32]. A study was conducted to analyze the release of drugs from the amnion. No difference was discovered between the analyzed soaking times and drug release. A longer drug soaking time actually reduces the effectiveness of drug uptake in the amnion. Based on the data included in the publications, 2–3 h of amnion soaking seem sufficient to fill up the membrane with medicines. However, it was revealed that the thickness of the amnion is very important in drug release. The entrapment efficiency of drugs was significantly higher in thicker amnions than in thinner amnions. In our study, we used the maximum amnion thickness which could be achieved in our tissue bank by soaking it in antibiotics for 3 h at a temperature of 4 °C. It was proven that the amnion may be an excellent carrier for not only stable drug formulations, but also fortified ophthalmic formulations, and it can release drugs for a period of 5 days without compromising stability. All of these data, as well as our studies, present a new clinical path for the use of the amnion not only as a skin substitute, but also as a medium for antiseptics, antibiotics, or ophthalmology drugs.

## 5. Conclusions

In our study, we demonstrated the ability of the amniotic membrane to inhibit the growth of bacteria isolated from burn wounds. The use of the amnion soaked in the saline solution, regardless of the method of preparation, inhibits the growth of bacteria. However, there is a significant impact of the method of preparing the amnion soaked in antiseptics on the degree of microbial inhibition. A much greater inhibition of microbial growth was demonstrated by the amnion sterilized by radiation and previously subjected to a cryopreservation procedure and soaked in antiseptic in comparison to the fresh amnion. Our research also indicates weak inhibition properties of microbial growth isolated from burn wounds by amnion soaked in Microdacyn regardless of the method of its preparation.

Our research indicates the potential for using the amniotic membrane for the treatment of chronic wounds, including infected wounds.

## Figures and Tables

**Figure 1 medicina-60-01015-f001:**
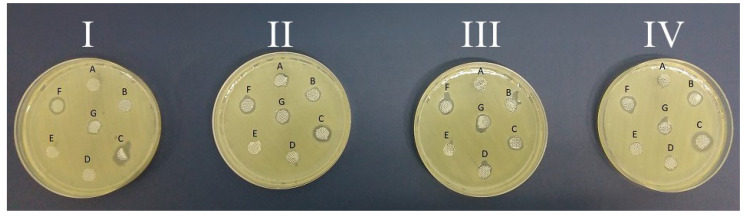
Photographs of the growth inhibition of *Pseudomonas aeruginosa* (Ps) at the sites of application of discs soaked with antiseptics: A—NaCl (control); B—Prontosan; C—Braunol; D—Borasol; E—Mircrodacyn; F—Octenilin; and G—Sutrisept on (**I**) fresh human amnion (biologically vital), (**II**) radiation-sterilized human amnion (biostatic), (**III**) fresh porcine amnion (biologically vital), and (**IV**) radiation-sterilized porcine amnion (biostatic).

**Figure 2 medicina-60-01015-f002:**
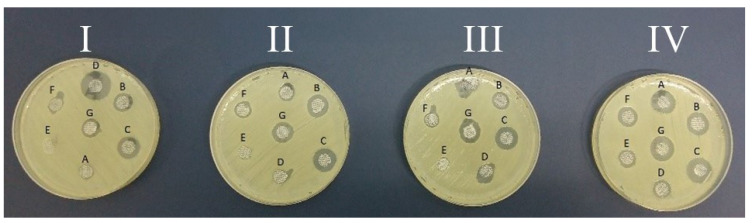
Photographs of the growth inhibition of *Acinetobacter baumannii* (Ac) at the sites of application of discs soaked with antiseptics: A—NaCl (control); B—Prontosan; C—Braunol; D—Borasol; E—Mircrodacyn; F—Octenilin; and G—Sutrisept on (**I**) fresh human amnion (biologically vital), (**II**) radiation-sterilised human amnion (biostatic), (**III**) fresh porcine amnion (biologically vital), and (**IV**) radiation-sterilized porcine amnion (biostatic).

**Figure 3 medicina-60-01015-f003:**
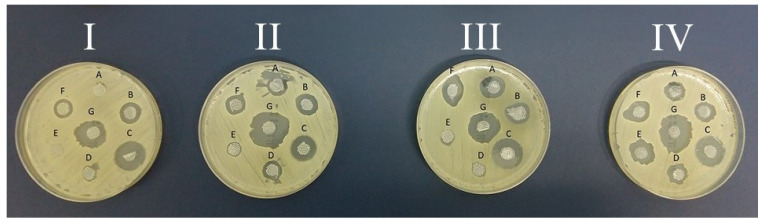
Photographs of the growth inhibition of *Methicillin-sensitive Staphylococcus aureus* (MSSA) at the sites of application of discs soaked with antiseptics: A—NaCl (control); B—Prontosan; C—Braunol; D—Borasol; E—Mircrodacyn; F—Octenilin; and G—Sutrisept on (**I**) fresh human amnion (biologically vital), (**II**) radiation-sterilized human amnion (biostatic), (**III**) fresh porcine amnion (biologically vital), and (**IV**) radiation-sterilized porcine amnion (biostatic).

**Figure 4 medicina-60-01015-f004:**
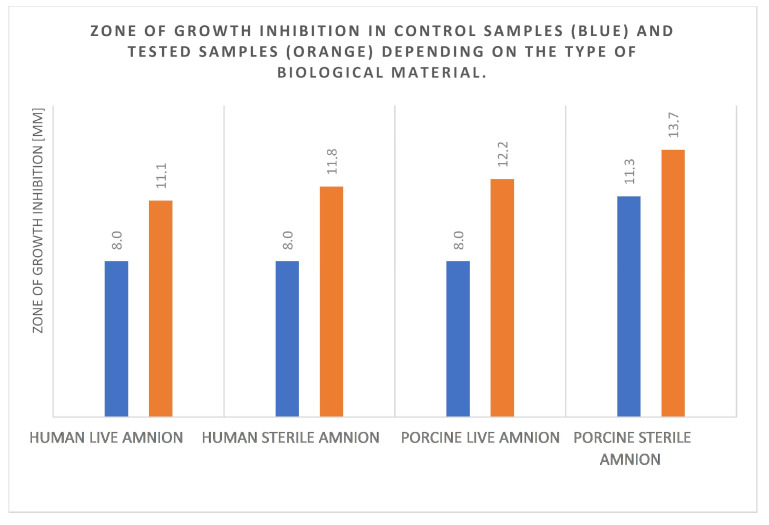
Bar chart of bacterial growth inhibition zones [mm] depending on type of amnion: live human, radiation-sterilized human, live porcine, and radiation-sterilized porcine amnions.

**Figure 5 medicina-60-01015-f005:**
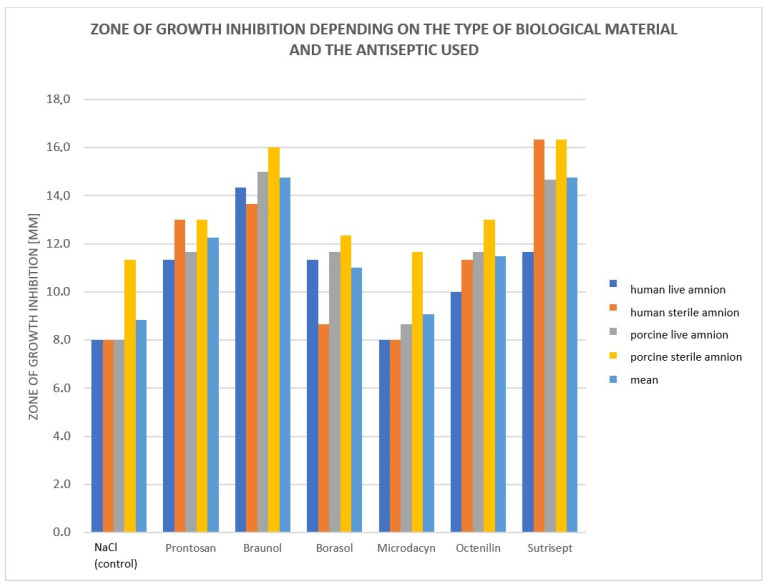
Bar chart of bacterial growth inhibition zones [mm] depending on type of antiseptic used in different kinds of amnion: fresh human, radiation-sterilized human, fresh porcine, and radiation-sterilized porcine amnions.

**Figure 6 medicina-60-01015-f006:**
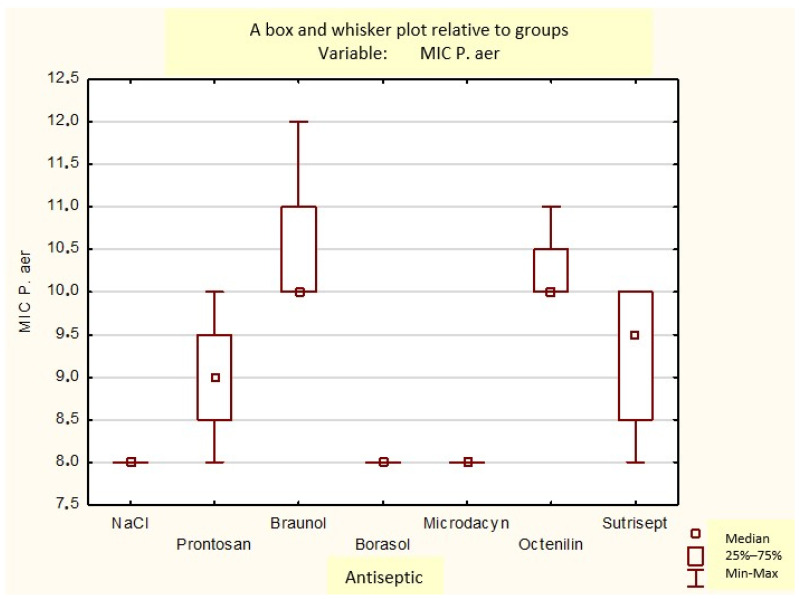
A box and whisker plot of the growth inhibition zones of *Pseudomonas aeruginosa* for the used antiseptics.

**Figure 7 medicina-60-01015-f007:**
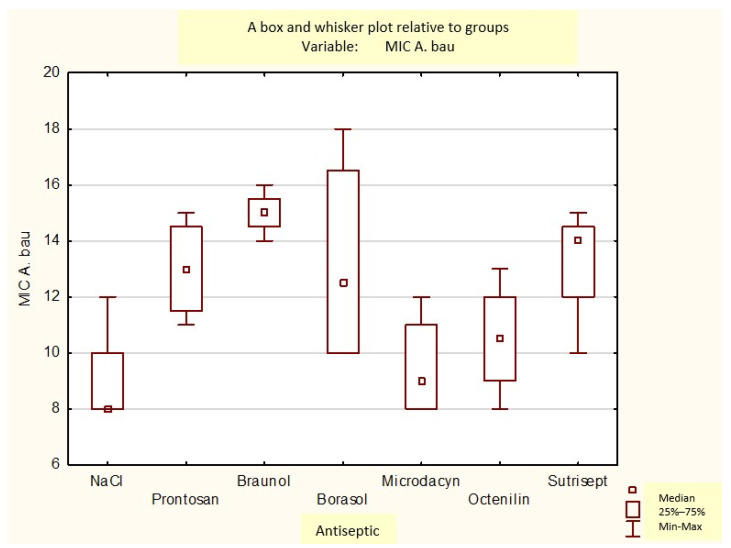
A box and whisker plot of the growth inhibition zones of *Acinetobacter baumannii* for the used antiseptics.

**Figure 8 medicina-60-01015-f008:**
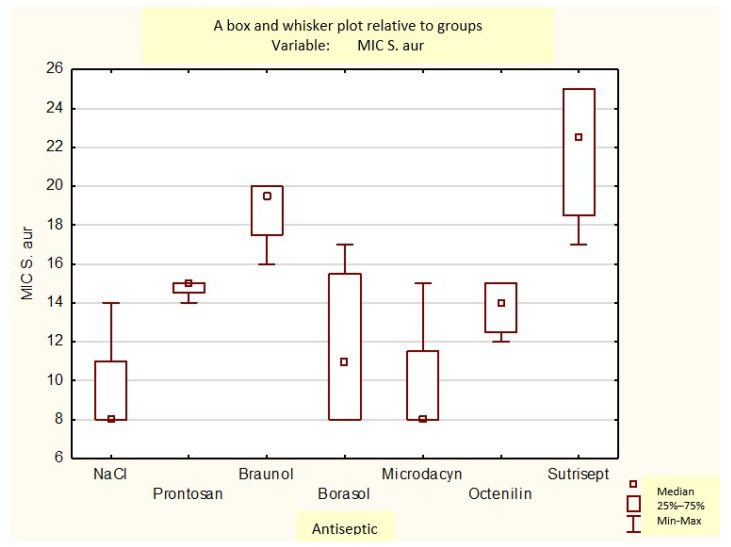
A box and whisker plot of the growth inhibition zones of methicillin-sensitive *Staphylococcus aureus* (MSSA) for the used antiseptics.

**Table 1 medicina-60-01015-t001:** Diameters of inhibited bacterial strain growth areas (mm) after application of fresh human amnion (biologically vital) impregnated with antiseptics.

	Antiseptic	A NaCl	B Prontosan	C Braunol	D Borasol	E Microdacyn	F Octenilin	G Sutrisept
Strain								
*Pseudomonas aeruginosa*		8	8	10	8	8	10	8
*Acinetobacter* *baumannii*		8	12	14	18	8	8	10
*Methicillin-sensitive Staphylococcus aureus (MSSA)*		8	14	19	8	8	12	17

**Table 2 medicina-60-01015-t002:** Diameters of inhibited bacterial strain growth areas (mm) after application of radiation-sterilized human amnion (biostatic) impregnated with antiseptics.

	Antiseptic	A NaCl	B Prontosan	C Braunol	D Borasol	E Microdacyn	F Octenilin	G Sutrisept
Strain								
*Pseudomonas aeruginosa*		8	10	10	8	8	10	10
*Acinetobacter baumannii*		8	14	15	10	8	11	14
*Methicillin-sensitive Staphylococcus aureus (MSSA)*		8	15	16	8	8	13	25

**Table 3 medicina-60-01015-t003:** Diameters of inhibited bacterial strain growth areas (mm) after application of fresh porcine amniotic membrane (biologically vital) impregnated with antiseptics.

	Antiseptic	A NaCl	B Prontosan	C Braunol	D Borasol	E Microdacyn	F Octenilin	G Sutrisept
Strain								
*Pseudomonas aeruginosa*		8	9	10	8	8	10	10
*Acinetobacter baumannii*		8	11	15	10	10	10	14
*Methicillin-sensitive Staphylococcus aureus (MSSA)*		8	15	20	17	8	15	20

**Table 4 medicina-60-01015-t004:** Diameters of inhibited bacterial strain growth areas (mm) after application of porcine amniotic membrane sterilized with radiation (biostatic) and impregnated with antiseptics.

	Antiseptic	A NaCl	B Prontosan	C Braunol	D Borasol	E Microdacyn	F Octenilin	G Sutrisept
Strain								
*Pseudomonas aeruginosa*		8	9	12	8	8	11	9
*Acinetobacter baumannii*		12	15	16	15	12	13	15
*Methicillin-sensitive Staphylococcus aureus (MSSA)*		14	15	20	14	15	15	25

## Data Availability

Data are contained within the article.

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
