# Peer review of "Amnion as an Innovative Antiseptic Carrier: A Comparison of the Efficacy of Allogeneic and Xenogeneic Transplantations in the Context of Burn Therapy"

_medicina, 2024, doi:10.3390/medicina60061015_

Round 1

Reviewer 1 Report

Comments and Suggestions for Authors

a. The most important comment for this study is about the main aim of the study. the point is here that the authors tried to compare two types of amnionic membranes by soaking them in various antiseptics. the point is here that those antiseptics have been approved previously and they have nothing new in wound treatment. Most importantly, it seems that the authors tried to use a carrier for better antiseptic delivery. if this is the main idea, so the authors forgat several factors to consider to choose the best carrier. So, in this regard, the authors must answer the following questions:

1. why did you choose amnionic membranes? there are other options by which not only you can provide more groups but also they are cost-effective and also more acceccible. 

2. if you aim to introduce a career, so why didn't you study more physical and mechanical properties? before going for any in vivo study or biological studies, it is mandatory to evaluate the membranes from the viewpoint of physical and mechanical properties. For example, parameters like degradation, swelling, mechanical strength, porosity, ... are necessary to evaluate a carrier.

3. it is mandatory to measure the amount of loaded antiseptics in each membrane. When the amount of loaded antiseptic is not equal in each membrane, it is not recommended to compare the results. So, a part of your research must focus on the potential of each membrane on antiseptic loading.

4. last but not least, considering carriers, the rate of drug-loaded release from them must be considered. 

I need the authors to address these comments first.

Comments on the Quality of English Language

a minor correction is recommended. 

Author Response

Thank you very much for taking the time to review this manuscript. Please find the detailed responses below:

Comments 1: Why did you choose amnionic membranes? there are other options by which not only you can provide more groups but also they are cost-effective and also more acceccible.

Response 1: The authors are employees of a Tissue Bank, which procures amniotic membrane and then prepares biovital grafts or biostatic dressings for the treatment of burn and chronic wound patients. The source of the fetal tissues is the placenta, which is delivered during the third stage of labor and is classified as biological waste in most medical facilities. Its availability is very high, with 3-4 cesarean section deliveries occurring daily in just one of the many hospitals with which agreements for tissue procurement can be made. Our Bank has agreements with three hospitals due to the personnel limitations of our production staff, who prepare the above- mentioned amniotic products. Preparing amniotic membrane is significantly more cost-effective for a hospital with a Tissue Bank than purchasing ready-made synthetic dressings. For comparison, the preparation cost of 1 cm² of amniotic membrane is 4 to 5 times cheaper than purchasing commercially available skin substitutes such as Suprathel, which is also used in our hospital.

Comments 2: If you aim to introduce a career, so why didn't you study more physical and mechanical properties? before going for any in vivo study or biological studies, it is mandatory to evaluate the membranes from the viewpoint of physical and mechanical properties. For example, parameters like degradation, swelling, mechanical strength, porosity, ... are necessary to evaluate a carrier.

Response 2: The aim of this study was to evaluate the use of the amniotic membrane as a carrier of antiseptics in the local therapy of wounds of various etiologies. In this application, the amniotic membrane serves as a temporary skin substitute. Its role can include protecting the wound from the adverse effects of the external environment, preventing fluid and electrolyte loss, and possibly stimulating natural regenerative processes. Therefore, the role of amniotic grafts in this context is not to act as a scaffold that physically withstands natural forces, as is the case with ADM grafts, pericardium, or fascia in certain fields of reconstructive surgery (e.g., hernias, procedures in female pelvic organ prolapse disorders). Consequently, parameters such as mechanical strength were not examined in this study.

It is important to note that there are numerous studies evaluating the physical parameters (e.g., strength) of amniotic membranes. However, based on the extensive clinical experience of our center in using allogeneic amniotic membrane grafts for treating burns, chronic wounds, and other skin diseases (e.g., Lyell's syndrome) with satisfactory clinical outcomes, the authors consciously focused on the aspect of delivering antibacterial substances through the amniotic membrane rather than on its physical parameters. The authors based their research planning on existing results to avoid duplicating studies that would only confirm the already proven physical properties of the amniotic membrane.

There are studies which indicate that the physical property of the amnion, having numerous internal spaces, allows them to be filled with liquid (antibiotic or antiseptic), which further increases the absorbency of the amniotic membrane [1]. It has also been demonstrated that active substances can diffuse through the amniotic membrane [2]. The permeability of fresh and cryopreserved amnion to antibiotics used topically in ophthalmology was tested in an in vitro model. It was shown that both fresh and cryopreserved amniotic membranes are permeable to aqueous solutions of ofloxacin [2]. The pharmacokinetic release of the drug from the amniotic membrane was quantitatively assessed, showing significant capacity of the amnion to act as a carrier for the slow release of the drug over a period of up to 7 hours in vitro. Ofloxacin was detectable in the acceptor phase after just 1 minute, with a rapid increase in drug release observed after two hours. These data should be considered in the clinical use of amnion as an antiseptic carrier [3]. These properties result from the physiological role of the placenta, in which amniotic fluid transports nutrients to the amnion by diffusion. The amniotic membrane itself is neither vascularized nor innervated, and nourishment occurs directly through diffusion from the amniotic fluid [4].

  1. Kobayashi A, Sugiyama K, Li W, Tseng SC. In vivo laser confocal microscopy findings of cryopreserved and fresh human amniotic membrane. Ophthalmic Surg Lasers Imaging. 2008 Jul-Aug;39(4):312-8. doi: 10.3928/15428877-20080701-10. PMID: 18717437.
  2. Resch MD1, Resch BE, Csizmazia E, Imre L, Németh J, Révész P, Csányi E. Permeability of human amniotic membrane to ofloxacin in vitro. Invest Ophthalmol Vis Sci. 2, 2010, Tom 51, 1024-7.
  3. Resch MD, Resch BE, Csizmazia E, Imre L, Németh J, Szabó-Révész P, Csányi E. Drug reservoir function of human amniotic membrane. J Ocul Pharmacol Ther. 2011 Aug;27(4):323-6. doi: 10.1089/jop.2011.0007. Epub 2011 Jun 24. PMID: 21702686.
  4. Gupta A, Kedige SD, Jain K. Amnion and Chorion Membranes: Potential Stem Cell Reservoir with Wide Applications in Periodontics. Int J Biomater. 2015;2015:274082. doi: 10.1155/2015/274082. Epub 2015 Dec 6. PMID: 26770199; PMCID: PMC4684856.

Comments 3: It is mandatory to measure the amount of loaded antiseptics in each membrane. When the amount of loaded antiseptic is not equal in each membrane, it is not recommended to compare the results. So, a part of your research must focus on the potential of each membrane on antiseptic loading.

Response 3: In the presented study, amniotic discs were prepared in a reproducible manner using a Disposable Biopsy Punch tool of the appropriate diameter, ensuring consistency in the size of the prepared samples. A multi-well plate, providing uniform well capacity, was used in the study, where amniotic discs of the same consistent diameter were placed. An identical volume of antiseptic agent was added to each well. Such experimental conditions ensure the same consistent amount of antiseptic agent in each tested sample. Each antiseptic was used to soak the same amniotic membrane, derived from the same donor or the same animal, which eliminates variability even between individual specimens. This experiment has clinical implications in hospital settings, particularly in surgical and chronic wound care departments, where due to equipment limitations, the amniotic membrane is soaked in an antiseptic and applied to infected wounds. Therefore, the amniotic grafts were soaked in specific antiseptic preparations, each possessing a unique, specified concentration of a particular substance. This was done according to a developed methodology. It should be assumed that each amniotic fragment reached its maximum saturation level with each tested substance, which was indirectly verified in microbiological studies.

Comments 4. Last but not least, considering carriers, the rate of drug-loaded release from them must be considered.

Response 4: The procedure for inhibiting the growth of microorganisms isolated from burn wounds of patients in our hospital was conducted in a certified microbiology laboratory, following validated and reproducible procedures. These studies involved testing the inhibition of microbial growth through the release of antibiotics into microbiological media (antibiogram). This procedure ensures the reproducibility of results regardless of the antibiotic used and is routinely employed in microbiology laboratories to assess the resistance of microorganisms isolated from patients and their sensitivity to a given antibiotic, allowing for targeted therapy selection. Our study was performed using the diagnostic method applied in clinical diagnostics by the same laboratory, confirming the clinical application of this antiseptic carrier to inhibit microbial growth.

We have attached the revised manuscript. We hope that our revisions and responses to the reviewer's comments will be satisfactory.

Sincerely yours

Agnieszka Klama-Baryła

Reviewer 2 Report

Comments and Suggestions for Authors

The article provides a thorough evaluation of the effectiveness of using amniotic membranes as an antiseptic vehicle in the context of burn therapy. The promise of this technique in inhibiting microbial growth and promoting wound healing is clearly highlighted. However, some suggestions could further improve the article:

Deepening clinical applicability: Integrate a section that discusses more fully the practical application of these findings in burn therapy. This could include considerations of safety, frequency, and duration of treatment with antiseptic-impregnated amniotic membranes.

Discussion of study limitations: It is important to discuss the limitations of the study, such as sample size or specificity of the bacterial strains tested, to provide a balanced assessment of the results.

Implications and future prospects: Adding a section on the future implications of this study and directions for further research could help to further contextualize the importance of the findings.

Author Response

Thank you very much for taking the time to review this manuscript. Please find the  responses below and the corresponding corrections highlighted in blue font in the re-submitted files.

Comments 1: Deepening clinical applicability: Integrate a section that discusses more fully the practical application of these findings in burn therapy. This could include considerations of safety, frequency, and duration of treatment with antiseptic-impregnated amniotic membranes.

Comments 2: Discussion of study limitations: It is important to discuss the limitations of the study, such as sample size or specificity of the bacterial strains tested, to provide a balanced assessment of the results.

Comments 3: Implications and future prospects: Adding a section on the future implications of this study and directions for further research could help to further contextualize the importance of the findings.

Response 1, 2, 3: We have extended the manuscript by expanding the Discussion section. Additionally, the Discussion now includes the information that the reviewer suggested to elaborate on. We have discussed more fully the practical application of these findings in burn therapy: safety, frequency, and duration of treatment with antiseptic-impregnated amniotic membranes. We have discussed the limitations of the study, such as sample size or specificity of the bacterial strains tested. We have presented implications and future prospects. All changes in the Discussion section have been marked in blue font. References 31 and 32 have been added.

We have attached the revised manuscript. We hope that our revisions and responses to the reviewer's comments will be satisfactory.

Sincerely yours

Agnieszka Klama-Baryła

Round 2

Reviewer 1 Report

Comments and Suggestions for Authors

Dear authors 

I was nearly persuaded with your answers for questions 1 and 2. But about the comments 3 and 4, you just explained some general information and you did not address my concerns. 

I think these two issues must be considered. 

Comments on the Quality of English Language

Just some typos mistakes 

Author Response

Thank you very much for taking the time to review this manuscript. Please find the detailed responses below:

Comments: I was nearly persuaded with your answers for questions 1 and 2. But about the comments 3 and 4, you just explained some general information and you did not address my concerns.

I think these two issues must be considered.

Comments 3. It is mandatory to measure the amount of loaded antiseptics in each membrane. When the amount of loaded antiseptic is not equal in each membrane, it is not recommended to compare the results. So, a part of your research must focus on the potential of each membrane on antiseptic loading.

Response 3: The amount of each antiseptic used in the study was measured at the time of its administration by taking a specified quantity of the antiseptic and adding it to a multi-well plate with the same volume per well in each trial. The amount of each antiseptic used in the study was measured prior to administration using a properly calibrated pipette, selected based on the specific volumes required in the study by choosing pipettes with precise liquid handling parameters. This ensured the comparability of all results by adding the same volume of antiseptic in each tested trial. Each antiseptic was applied to the same amniotic membrane, derived from the same donor or the same animal, eliminating variability even at the individual level. The amniotic membrane in each trial was cut to the same size using a validated tool with appropriate parameters, specifically a Disposable Biopsy Punch, which is registered as a medical device for clinical and diagnostic use. The antiseptics used in the study were purchased from the manufacturer. They are commercially available in Europe and have the necessary studies confirming their registration and CE marking as medical devices for clinical use.

Comments 4. Last but not least, considering carriers, the rate of drug-loaded release from them must be considered.

Response 4: The authors are employees of a tissue bank in a burn-specialized hospital that prepares tissue transplants for transplantation. They have limited research facilities, relying solely on collaborating laboratories such as the microbiology lab. The results obtained from the conducted experiments were used for the clinical application of the amnion as a carrier for antiseptics in the surgical, burn, and wound care departments of the authors' hospital. The pilot application of these studies demonstrated the remarkable effectiveness of amnion soaked with antiseptics in treating infected wounds. This indicates good release of antiseptics into the wound from the amnion over the appropriate time, up to 24 hours, which confirmed the clinical application of this method. The simplicity of soaking the amnion in an antiseptic solution allows for its widespread use. Available literature suggests that many hospitals face similar challenges in combating infections. The dissemination of this method by demonstrating its effectiveness, even at the laboratory research level, could lead to interest in this treatment approach in other facilities. The authors plan to continue their research in the future by collaborating with scientific institutions to investigate more thoroughly the release dynamics of antiseptics from the amnion. Currently, however, they do not have this capability. With this article, the authors aim to begin a series of publications to also demonstrate the clinical effectiveness of the amnion as an antiseptic carrier and to examine the parameters of the amnion, including the rate of antiseptic release, which was not possible due to their limited research facilities, if they can establish collaborations with research institutions capable of studying these parameters.

Therefore, this reviewer’s comment is particularly valuable as it highlights an important issue that the authors had not addressed in their research. Thanks to this, in the future, they will focus on this problem and aim to explore the rate of antiseptic release into the wound, studying it both in the laboratory and in the clinic.

We hope that responses to the reviewer's comments will be satisfactory.

Sincerely yours

Agnieszka Klama-Baryła

Round 3

Reviewer 1 Report

Comments and Suggestions for Authors

Actually, I need numbers about the percentage of loaded antiseptic and antiseptic release rate.

Again you explained some general information not addressing the exact numbers.

At this stage I do understand that your research is finished and maybe you cannot perform new tests .

However, please consider these parameters whenever you are working on loading a kind of pharmaceutical agent on a carrier.

With consideration and regarding the next step of your research I accept your responses.